# Feasibility and Efficiency of the BEFORE (Better Exercise and Food, Better Recovery) Prehabilitation Program

**DOI:** 10.3390/nu13103493

**Published:** 2021-10-02

**Authors:** Thaís T. T. Tweed, Misha A. T. Sier, Ad A. Van Bodegraven, Noémi C. Van Nie, Walther M. W. H. Sipers, Evert-Jan G. Boerma, Jan H. M. B. Stoot

**Affiliations:** 1Department of Surgery, Zuyderland Medical Center, Dr. H van der Hoffplein 1, 6162BG Sittard-Geleen, The Netherlands; m.sier@zuyderland.nl (M.A.T.S.); no.visser@zuyderland.nl (N.C.V.N.); e.boerma@zuyderland.nl (E.-J.G.B.); j.stoot@zuyderland.nl (J.H.M.B.S.); 2Department of Gastroenterology, Zuyderland Medical Center, Dr. H van der Hoffplein 1, 6162BG Sittard-Geleen, The Netherlands; a.vanbodegraven@zuyderland.nl; 3Department of Geriatric Medicine, Zuyderland Medical Center, Dr. H van der Hoffplein 1, 6162BG Sittard-Geleen, The Netherlands; w.sipers@zuyderland.nl

**Keywords:** prehabilitation, colorectal surgery, multimodal, functional capacity, enhanced recovery after surgery, complications, colorectal cancer, feasibility

## Abstract

Prehabilitation has been postulated as an effective preventive intervention to reduce postoperative complications, particularly for elderly patients with a relatively high risk of complications. To date, it remains to be determined whether prehabilitation increases physical capacity and reduces postoperative complications. The aim of this study was to assess the feasibility of a 4-week multimodal prehabilitation program consisting of a personalized, supervised training program and nutritional intervention with daily fresh protein-rich food for colorectal cancer patients aged over 64 years prior to surgery. The primary outcome was the feasibility of this prehabilitation program defined as ≥80% compliance with the exercise training program and nutritional intervention. The secondary outcomes were the organizational feasibility and acceptability of the prehabilitation program. A compliance rate of ≥80% to both the exercise and nutritional intervention was accomplished by 6 patients (66.7%). Attendance of ≥80% at all 12 training sessions was achieved by 7 patients (77.8%); all patients (100%) attended ≥80% of the available training sessions. Overall, compliance with the training was 91.7%. Six patients (66.7%) accomplished compliance of ≥80% with the nutritional program. The median protein intake was 1.2 (g/kg/d). No adverse events occurred. This multimodal prehabilitation program was feasible for the majority of patients.

## 1. Introduction

Despite optimization of surgical techniques with the introduction of minimally invasive surgery and the improvement of perioperative care with Enhanced Recovery After Surgery (ERAS) programs, postoperative complications after colorectal surgery still occur in approximately one-third of patients undergoing colorectal resection, with the elderly at increased risk [1,2,3]. Worldwide, colorectal cancer is the third most common cancer and second leading cause of cancer death. Annually, approximately 2 million patients are diagnosed with colorectal cancer, of which the majority occurs in the increasing older adult population [4,5,6,7]. Since surgery is the standard curative treatment for colorectal cancer, further reduction of complications is paramount [8]. Recently, interest in reducing complications by enhancing patients’ condition prior to surgery is rising. Factors that adversely affect outcomes include co-morbidity, polypharmacy, cognitive impairment, dependency, and frailty, which can be measured using the preoperative comprehensive geriatric assessment (CGA) [9,10]. Frailty is characterized by age-related deviations that lead to decreased energy and muscle strength, weight loss, and sedentary activity levels [11,12].

In previous studies it has been suggested that some factors adversely affecting postoperative outcomes may be modified before surgery [13]. Prehabilitation aims to increase physiological reserve in anticipation of adverse effects of surgery and to optimize postoperative recovery, especially in the frail patient population [14,15,16]. It has been postulated that prehabilitation, including nutritional support or physical exercise training prior to surgery, could be effective in reducing patient frailty and improving outcomes after abdominal surgery [13,15,16,17,18,19,20,21,22].

The beneficial impact of prehabilitation on physical capacity and postoperative outcomes remains inconclusive due to contradictory results, small study samples, and large heterogeneity [13,18,19,20,21,22,23,24,25]. This heterogeneity encompasses variation in the content of the prehabilitation program. Nonetheless, in several studies the effects of multimodal prehabilitation consisting of physical training in combination with nutritional support using oral nutritional supplements (ONS), e.g., Nutridrink^®^ have been explored [26,27]. To our knowledge, no trials have been conducted to determine the effect of a multimodal prehabilitation program for elderly patients prior to elective colorectal resection for malignancies that combined personalized physical exercise training and fresh protein-rich food.

The aim of this study was to explore the feasibility of the BEFORE (Better Exercise and Food, Better Recovery) multimodal prehabilitation program consisting of personalized, ambulatory, hospital based exercise training, and fresh protein-rich food in terms of compliance, organization and acceptance to outline the design of a large, statistically well-powered comparative trial.

## 2. Materials and Methods

### 2.1. Study Design

This prospective feasibility study was conducted in one large Dutch teaching hospital (Zuyderland Medical Center, Sittard-Geleen, The Netherlands). After detection of a colorectal malignancy during endoscopy, patients were assessed for eligibility and were enrolled at the outpatient clinic between November 2019 and March 2020. Data were collected at baseline, during all training sessions, before surgery and at 30-days, and 1-year follow-up.

### 2.2. Study Participants

Patients with a colorectal (pre)malignancy were recruited if they were scheduled for an elective colorectal resection and if they met the following criteria: age > 64 years, BMI < 35 kg/m^2^, physically and mentally capable of completing the exercise program, the ability to answer questionnaires and the ability to orally consume (calculated) daily nutritional requirements. The exclusion criteria were parenteral nutrition or enteral nutrition via feeding tube in the preoperative phase, a history of or an active inflammatory gastrointestinal disease, a palliative treatment course, previous participation in a multimodality approach study, inability or contraindication to exercise, intellectual disability, complex dietary needs or food allergy.

Ethical approval was obtained from the local Medical Ethics Review Committee Zuyderland: Medisch-Etische Toetsingscommissie Zuyderland (METC Z), (NL70834.096.19) the study was conducted according to the ethical standards of the Helsinki Declaration of 1975. All patients provided written informed consent.

### 2.3. Baseline Assessment

Baseline assessment was performed within one week after enrollment by the Sport and Exercise Physician (SEP) and an in-hospital dietician, prior to the start of the multimodal prehabilitation program. A second assessment was conducted after completion of the BEFORE 4-week multimodal prehabilitation program, prior to surgery.

The initial physical screening included physical examination, rest-electrocardiogram (ECG), rest-spirometry and Cardiopulmonary Exercise Testing (CPET; Quark CPET, Cosmed, Rome, Italy) on a calibrated electronically braked cycle ergometer (Excalibur Sport, Lode B.V., Groningen, the Netherlands). To assess baseline aerobic fitness and to establish personalized intensities for the supervised training program, CPET determined heart rate, maximum oxygen uptake (VO_2_max), ventilatory anaerobic threshold (VAT), respiratory compensation point (RCP), carbon dioxide production, respiratory flow and volume parameters in the breath-by-breath measurements. Muscle strength was measured by testing handgrip strength of the dominant hand in 3 consecutive cycles, of which the mean score was determined. To assess nutritional requirements, the dietician asked participants about their daily food consumption, based on which the current protein and energy intake were estimated. In accordance with the European Society for Clinical Nutrition and Metabolism (ESPEN) guidelines, the required protein intake was set at 1.2 to 1.5g protein/kg bodyweight [28]. The required energy intake was calculated using the Harris-Benedict formula with an addition of 30% to correct for the malignant disease [29].

Further descriptive assessment included the Short Nutritional Assessment Questionnaire (SNAQ) [30], the Groningen Frailty Index (GFI) [31], the European Organization for Research and Treatment for Cancer Quality of Life Questionnaire (EORTC QLQC-30) [32], and a VAS [33] questionnaire, designed for this study, to evaluate appetite and food experience (see Appendix B).

The reassessment included physical examination, CPET, dietary reassessment and a questionnaire evaluating patients’ experience with the multimodal prehabilitation program (see Appendix A).


BEFORE prehabilitation program


The BEFORE prehabilitation program has been specifically developed for this study and consisted of an exercise and nutritional intervention.

### 2.4. Exercise Intervention

During four weeks, between diagnosis and surgery, patients received personalized, supervised (by a specialized physiotherapist) exercise training combined with freshly prepared protein-rich food.

The ambulatory training sessions were organized in the hospital. The program consisted of training sessions of 60 to 75 min, three times a week for four weeks, including strength training followed by aerobic training. Strength training involved six functional upper and lower body push-pull exercises (deadlift, chest press, lateral pull down, leg press, shoulder press and seated row). During the first session, the personal one-repetition maximum (1-RM) of each exercise was determined for further training. Each exercise started with 20 repetitions (60–65% 1-RM), followed by 2 series with 6 repetitions (80–85% 1-RM). Aerobic training consisted of High Intensity Interval Training (HIIT) on the cycle ergometer with 30-s and 60-s intervals and a 1:3 work-recovery ratio, based on the personal ventilatory thresholds measured with CPET [34,35]. Compliance with the exercise program was determined by means of attendance, the mean duration of participation in the training program and the average achieved training intensity.

During all training sessions, the resistance (in kilograms) of the strength training was reported for each exercise. Similarly, the aerobic function was assessed during each session by measuring maximum heart rate and intensity using the Borg Scale [36].

### 2.5. Nutritional Intervention

Patients received three freshly prepared protein-rich meals (breakfast, lunch, dinner) and three snacks per day provided by Daily Fresh Food Inc.^®^, Geleen, the Netherlands. Daily Fresh Food Inc.^®^ produces and supplies high-quality, safe and fresh food. The European Union has assigned the following numbers to Daily Fresh Food: EG-454-NL, EG-923-NL. The nutrients contained the required amount of protein and calories as calculated by the dietician. To meet these requirements, the meals were prepared with additional fresh, protein-rich ingredients (a section of the menu is shown in Appendix C). Patients were not allowed to consume other nutrition. To accurately measure the macronutrient intake, patients were asked to complete a 7-day food diary, to weigh each dish and to take pictures of the meals before and after consumption with a provided Samsung tablet (Samsung Electronics Co., Ltd., Yeongtong District, Suwon, Korea). Daily Fresh Food Inc.^®^ provided a list of the nutritional value per meal. The macronutrient intake was calculated based on the measurements and nutritional list; unclear (vague or undistinguishable) pictures were reported as missing data.

### 2.6. Outcomes

The primary study outcome was the feasibility of the BEFORE multimodal prehabilitation program, defined as at least 80% adherence to the exercise training program and the nutritional intervention.

Secondary outcome measures were the organizational feasibility, the acceptability of the interventions, functional capacity after prehabilitation (determined with CPET measures), and muscle strength (determined with handgrip strength and 1 RM). Other secondary outcomes were length of hospital stay, postoperative complications, readmission rate, and 30-day and 1-year mortality. Complications were divided into surgical and non-surgical complications and scored according to the Clavien-Dindo classification [37,38].

The organizational feasibility was determined by calculating the success rate of meal order and delivery, as well as the success rate of organizing the intended supervised exercise training.

The acceptability of the nutritional and the exercise training intervention was measured by evaluating patients’ experiences with the nutritional intervention and the exercise training intervention using a questionnaire (see Appendix A). Likewise, we explored the feasibility of measuring food intake, and adjusting nutrition to the needs of patients and personalizing training.

### 2.7. Statistical Analysis

To study the feasibility of this multimodal prehabilitation program, the aim was to include 10 patients. As this is a feasibility study, no sample size or power analysis was performed.

The feasibility of the multimodal prehabilitation program was assessed using descriptive analysis. Continuous variables were presented as mean, median, percentage and frequency with standard deviations (SD). Categorical variables were presented as means, median, percentages, numbers and frequency with standard deviations (SD). The Wilcoxon signed rank test was used to assess change in physical capacity before and after prehabilitation. Data were analyzed with the Statistical Package for the Social Sciences for Windows (version 23.0; IBM, SPSS Inc., Chicago, IL, USA).

## 3. Results

Between the 1st of November 2019 and the 5th of March 2020, a total of 119 patients were assessed for eligibility for participation. On 5 March, the inclusion of patients was discontinued prematurely due to the Corona pandemic. Of this initial sample, 30 participants met the inclusion criteria. One patient (3.0%) was excluded based on a previous colorectal tumor resection, and 20 (66.7%) patients declined participation. The remaining 9 patients (30.0%) consented to participate. A consort diagram for the study is presented below (see Figure 1).


Patient Characteristics


The baseline characteristics of the study participants are shown in Table 1. The median age of the participants was 73.0 (IQR 70.0–76.0), and slightly more participants were male (*n* = 5, 55.6%). Two patients (22.2%) had an ASA score of III. Four (44.4%) patients were classified as frail based on a GFI score ≥ 4. Moreover, one patient (11.1%) was classified as having a high nutritional risk.


Primary outcome


A compliance rate above 80% to both the exercise training and nutritional intervention was established by 6 patients (66.7%). The exercise and nutritional data are shown in Table 2.

### 3.1. Adherence Prehabilitation

Seven patients (77.8%) attended ≥80% of all 12 training; all patients (100%) attended more than 80% of the available training sessions. Individual attendance to the 12 training sessions ranged from 75.0–95.8% with a median of 91.7%, and 2 patients (22.2%) attended all sessions. The reason for missing training sessions was early surgery; the 2 patients who did not attend 80% of the 12 sessions were both operated before the end of the prehabilitation program. One patient attended 8 out of 8 sessions, and the other patient attended 6 out of 6 sessions. The median number of sessions attended was 11 (IQR 9–11.5), and the median prehabilitation period was 26 days.

A total of 6 patients (66.7%) consumed more than 80% of the required protein intake, and all participants (100%) consumed more than 80% of the required energy intake. The median percentage of consumed meals and snacks was 95.2% (IQR 79.8–97.6%), and one patient consumed all meals and snacks. The median protein intake (g/kg/day) was 1.2 (IQR 1.1–1.5), and the median energy intake was 2417.3 kcal per day (IQR 1898.9–2646.3). All participants (100%) only consumed food provided by Daily Fresh Food; no other nutriments were consumed. No adverse effects related to prehabilitation were reported.


Secondary outcomes


### 3.2. Organizational Feasibility

Meals were delivered 3–4 times per week in the hospital and were provided during the training sessions. Overall, food was delivered according to schedule in all but one case (98.7%), this delivery was postponed for one day due to a roadblock.

### 3.3. Acceptability of the Interventions

The majority of the patients had a positive food experience; 77.8% rated the taste and quality as good or excellent. The number of meals was considered a lot (*n* = 6, 66.7%) or too much (*n* = 2, 22.2%), and meal portions were considered sufficient (*n* = 7, 77.8%).

The training program was appraised as good or excellent by all patients; most patients rated the intensity as sufficient (*n* = 4, 44.4%) or heavy (*n* = 4, 44.4%). The number of training sessions was appropriate (*n* = 9, 100%), and the length of sessions was sufficient (*n* = 7, 77.8%).

Generally, patients accepted the extra hospital visits (*n* = 5, 55.5%) but four patients considered the extra visits burdensome.

### 3.4. Physical Outcome Measures

Descriptive statistics for the physical outcomes are presented in Table 3. Figure 2, Figure 3, Figure 4 and Figure 5 show the physical measurements of all patients before and after prehabilitation. After prehabilitation, a median improvement of + 2 kg (IQR 1.1ߝ2.0) in handgrip strength (kg) was observed (*p* = 0.02, Z = −2.41) and a median improvement of +13 Watt (IQR 6.0–27.0) was observed in exercise capacity *(p* = 0.02, Z = −2.37). No difference was observed in the median maximum oxygen uptake (VO_2_ max) before and after prehabilitation; 16.25 (IQR 13.18–24.18) versus 17.55 (IQR 12.95–21.23), nor was a difference observed in median oxygen uptake at Ventilatory Anaerobic Threshold (VAT); 12.60 (IQR 9.65–17.15) versus 13.55 (IQR 9.68–17.55). Muscle strength, measured in one-repetition maximum (1 RM) of lower and upper extremity musculature, had increased during the program in all patients.

### 3.5. Surgical Outcomes

Table 4 provides an overview of surgical outcomes. Surgery was performed in six patients. Three participants were not operated as one patient was diagnosed with liver metastasis requiring neoadjuvant therapy, one patient had a complete response to the neoadjuvant therapy, and one patient was diagnosed with a benign lesion for which no operation was required (these decisions were made after inclusion). Complications occurred in 2 out of 6 patients (33%). One minor complication (tachycardia) occurred (for which a beta blocker was prescribed), and one severe complication (iatrogenic small bowel perforation) occurred, which required reoperation. Upon one year follow-up, no mortality was registered.

## 4. Discussion

The present study is, to our knowledge, the first prospective study specifically focusing on the feasibility of the BEFORE multimodal prehabilitation program that combined a personalized and supervised exercise training program (to enhance physical capacity) with freshly prepared protein-rich meals (to support the required protein and energy intake), for patients 65 years and older with colorectal malignancy scheduled for elective surgery.

This study showed that this multimodal prehabilitation program was feasible for the majority of the elderly patients included in this study. The organization of the program was found to be feasible and the interventions acceptable for the population. We observed a small increase in exercise capacity and muscular strength. No adverse events directly related to the prehabilitation program were reported.

The high adherence rate to this multimodal program reflects the reports of Carli et al. and Van Rooijen et al. [24,39], who reported a prehabilitation attendance rate of at least 80%. Other reported compliance rates to multimodal prehabilitation programs prior to elective abdominal cancer surgery ranged from 59 to 98 per cent [21].

On the other hand, the recruitment rate in this study was low. This was in accordance with several prior previous studies [40,41]. The most common reason for refusal of participation in this study was that the program was considered to be too demanding. This may be due to the fact that a recent diagnosis of cancer with forthcoming surgery negatively impacts the incentive to participate in multimodal prehabilitation programs. Furthermore, difficulty to travel to the hospital (e.g., dependency of transportation) or patients’ inability to combine the prehabilitation program with the role of informal caregiver (e.g., to spouse or family member), affected their willingness to participate. It has been previously noted that, in the older adult population, the number needed to screen for inclusion is 1:3 [42]. In contrast, the recruitment rate in studies assessing multimodal prehabilitation with home-based exercise training or a shorter prehabilitation program was higher [15,20,43]. This suggests that the recruitment rate of this multimodal prehabilitation program could be increased by organizing training sessions in the near residential area; preferably supervised home-based or community-based exercise training. The supervised exercise training sessions was considered a specific strength of this study, since supervision increases adherence to exercise programs [44].

The organization of training sessions in the outpatient setting was uncomplicated; all intended sessions were organized by a dedicated team. The Dutch guidelines state that treatment of colorectal cancer has to be effected within 7 weeks after diagnosis [45], generally surgery is performed within 5 weeks following diagnosis [46]. The oncological outcome does not improve when colorectal patients are operated within these 5 weeks [47]. In this study, all patients were operated within this time window, some were operated before the end of the prehabilitation program. In one case this was due to the COVID-19 pandemic, and the other operation was rescheduled due to postponement of another operation. Even though a 4-week prehabilitation program may be feasible, this raises the question whether this prehabilitation program can be realized for all patients, since time between diagnosis and operation might be less than 4 weeks for various reasons.

In general, the overall success rate of food delivery was high; only one delivery was postponed. Nonetheless, arranging food delivery on the training days was rather complex. As patients entered the prehabilitation program on different moments, providing patients with the correct meal boxes required careful and detailed organization. An alternative could be home delivery, which is easier to organize.

### Acceptability of Interventions

In general, participants considered the interventions to be acceptable in terms of frequency and intensity of physical training, and regarding number and taste of the meals.

In contrast to the majority of literature focusing on nutritional support by consultation of a dietician and/or oral nutritional supplements, in this study exercise training and 6 freshly prepared meals for 4 weeks were combined. Because the taste of oral nutritional supplements (ONS) may be disliked [48], this study aimed to increase adherence to the nutritional intervention and to improve food experience. Positively, the adherence to the nutritional program was high (95.2%). Of the three modalities used to accurately calculate the macronutrient intake, the food diary appeared to be the most feasible and accessible method. Adherence to the nutritional intervention was consistent with prior reports [26,27]. In these studies a food diary or interview was applied to assess compliance, no objective measurements were included.

Testing physical capacity using CPET was complicated for two patients (22.2%) due to cycling difficulties: one patient was unable to cycle due to balance disorder and one patient could only cycle with low resistance because of an artificial knee. In follow-up studies, other modalities to test vital capacity could be considered.

Due to the small study sample, no conclusion can be drawn regarding the correlation between prehabilitation and both functional and surgical outcomes.

Improvement of some aspects of the physical capacity, including exercise capacity, muscular strength, and a minor increase in FVC and FEV1 were observed during this prehabilitation program. However, the improvement in handgrip strength was below the minimally clinically important difference of at least 5 kg as described in the literature [49,50]. There is a lack of evidence of the minimal clinically important difference for the other parameters.

Increase in exercise capacity was also reported by Barberan et al. and Liu et al., demonstrating a significant improvement of aerobic capacity [51,52]. Increase in FVC was consistent with the study of Liu et al., who demonstrated an improvement in FVC after two weeks of multimodal prehabilitation [52]. Reported enhancement in peak VO_2_ and VAT was not corroborated by the findings of this study [53]. A possible explanation for this difference might be that the patients included in this study were generally in better physical shape than patients in previous studies, characterized by relatively low ASA score, and rather high VO_2_max and VAT. Increase in muscular strength was consistent with previous studies [39,53].

The overall complication rate was in accordance with a few studies [15,16], but not all prior studies [24,53], potentially due to difference in age and in other patient characteristics.


Limitations


This study was limited by the low recruitment rate. This could result in selection bias; patients with affinity for training could be more inclined to participate in a prehabilitation program. This may have affected validity and generalizability of study findings. Since this was a feasibility study with a small study population no conclusions regarding the effect of prehabilitation on functional and surgical outcomes could be drawn. However, the aim of this study was to assess the feasibility of the prehabilitation program, this could be determined.

Baseline information of non-participating patients was not collected in this study; therefore no comparison between these patient-groups can be accomplished. Nonetheless, reasons for refusal were collected and will be taken into account in subsequent studies.

Furthermore, measuring all nutritional intake for one week appeared to be complex. The three measurement methods (weighing, taking pictures and keeping a food diary) had to be performed frequently, for some participants it proved to be difficult to perform all these measurements correctly. Some patients forgot to determine their food intake, various pictures were blurred, and keeping a food diary was demanding. The results regarding intake presented in this study therefore may underestimate the real nutritional intake. However, unlike previous studies, in this study intake was objectively measured, yielding a more accurate result than subjective measurements [26,27].

Although patients included in this study considered the program to be acceptable, recruiting patients proved difficult. To optimize the feasibility of a study like this, efforts should be made to ensure that all eligible patients are willing and able to participate in the prehabilitation program. The recruitment pathway should be further investigated, preferably including qualitative analysis of considerations regarding participation. Moreover, opportunities of prehabilitation with supervised community- or homebased exercise training must be further investigated. In addition to CPET, reliability and tolerance of measuring physical function with other tests should be investigated. The effects of this personalized multimodal prehabilitation program may be explored on a larger scale to analyze and corroborate the effect on functional capacity and postoperative outcomes. A pilot study was designed to further assess the effects of this prehabilitation program, but its start has been postponed until after the Corona pandemic.

## 5. Conclusions

In this study it has been demonstrated that personalized multimodal prehabilitation with supervised exercise training combined with protein-rich meals prior to colorectal resection for (pre)malignancy is feasible for at least two-third of the participants. Most patients completed >80% of the prehabilitation program. The program was also feasible in terms of organization and application for participants. Improvement in some aspects of physical capacity was suggested. In order to improve recruitment rate, modifications for a definitive trial should be considered and implemented, whilst statistical power may be calculated with the demonstrated effects of multimodal prehabilitation programs.

## Figures and Tables

**Figure 1 nutrients-13-03493-f001:**
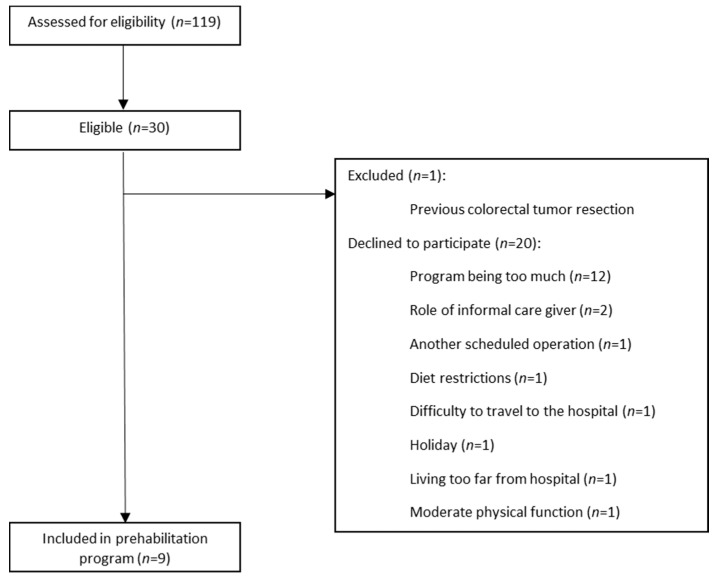
Consort diagram.

**Figure 2 nutrients-13-03493-f002:**
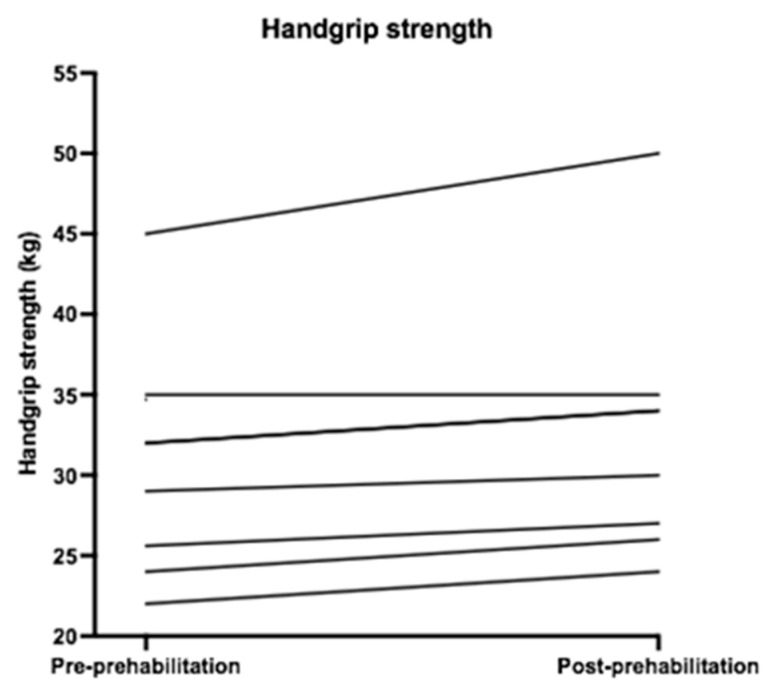
Handgrip strength (kg) before and after prehabilitation.

**Figure 3 nutrients-13-03493-f003:**
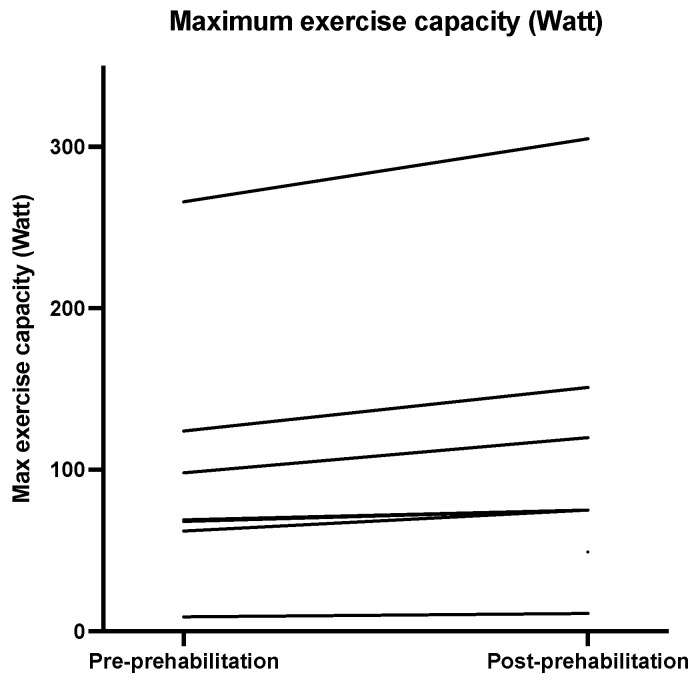
Maximum exercise capacity (Watt) before and after prehabilitation.

**Figure 4 nutrients-13-03493-f004:**
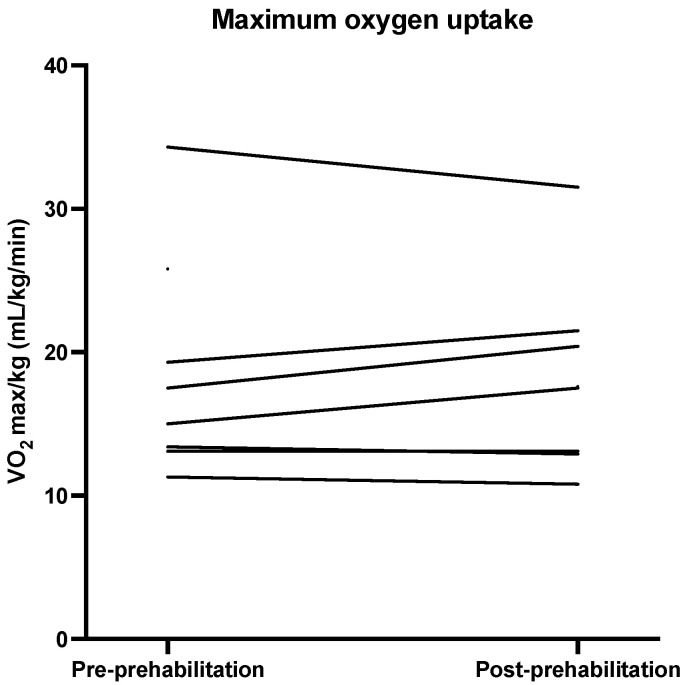
Maximum oxygen uptake (mL/kg/min) before and after prehabilitation.

**Figure 5 nutrients-13-03493-f005:**
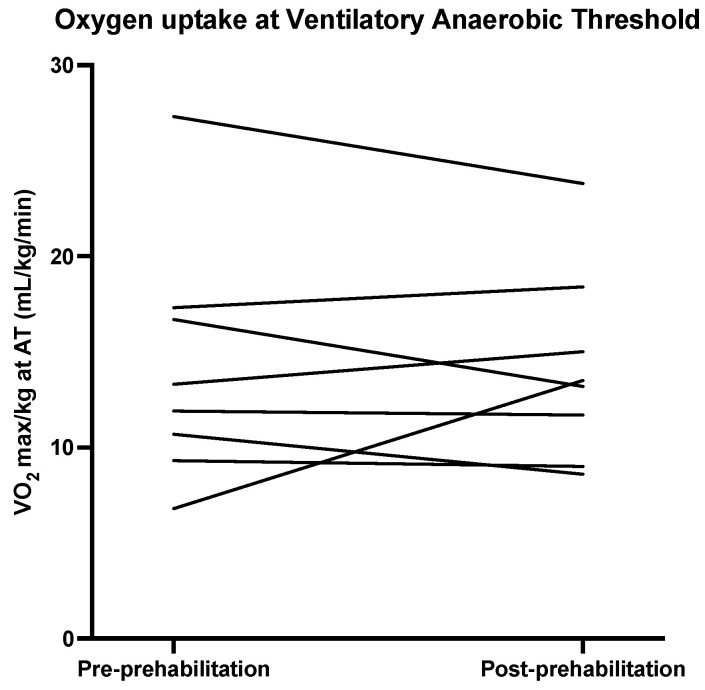
Oxygen uptake at Ventilatory Anaerobic Threshold (VAT) before and after prehabilitation.

**Table 1 nutrients-13-03493-t001:** Descriptive data for patients; median (interquartile range).

	Study Population (*n* = 9)
Age (years)	73.0 (70.0–76.0)
Gender ratio (M:F)	5:4
Height (cm)	169.0 (160.5–178.0)
Bodyweight (kg)	84.0 (73.5–91.4)
BMI ^1^ (kg/m^2^)	26.9 (25.0–32.7)
Smoking; *n* (%)	1 (11.1)
Alcohol; *n* (%)	1 (11.1)
Comorbidity; *n* (%)	
Cardiovascular disease	7 (77.8)
Pulmonary disease	5 (55.6)
History of abdominal surgery	2 (22.2)
History of other malignancy	2 (22.2)
ASA score ^2^; *n* (%)	
II	7 (77.8)
III	2 (22.2)
GFI ^3^; *n* (%)	
0	1 (11.1)
I	2 (22.2)
II	1 (11.1)
III	1 (11.1)
IV	2 (22.2)
V	2 (22.2)
SNAQ ^4^; *n* (%)	
0	6 (66.7)
I	2 (22.2)
V	1 (11.1)
Diagnosis; n	
Malignancy	8
Liver metastasis at diagnosis	1
Premalignancy	1
Adjuvant treatment; n	
Neoadjuvant chemoradiation	2
Adjuvant chemotherapy	1

^1^ BMI: body mass index; ^2^ ASA: American Society of Anesthesiologists physical status classification system; ^3^ GFI: Geriatric Frailty Index, ^4^ SNAQ: Short Nutritional Assessment Questionnaire.

**Table 2 nutrients-13-03493-t002:** Exercise and nutritional data for patients; *n* (%), median (interquartile range).

	Study Population (*n* = 9)
>80% of the prehabilitation program completed	6 (66.7)
>80% of all 12 training sessions attended; *n* (%)	7 (77.8)
>80% of all available training sessions attended; *n* (%)	9 (100)
Individual adherence percentage training sessions	91.7 (75.0–95.8)
Number of attended training sessions	11.0 (9.0–11.5)
Prehabilitation period (days)	26.0 (22.0–27.0)
>80% of the required protein consumed; *n* (%)	6 (66.7)
>80% of the required calories consumed; *n* (%)	9 (100)
Individual percentage meals consumed	95.2 (79.8–97.6)
Total protein intake (grams/day)	113.4 (87.1–114.1)
Protein intake (grams/kg bodyweight/day)	1.2 (1.1–1.5)
Total energy intake per day (kcal)	2417.3 (1898.9–2646.3)

**Table 3 nutrients-13-03493-t003:** Physical outcomes; median (interquartile range).

	Pre-Prehabilitation	Post-Prehabilitation	Difference	*p*-Value	Z-Value
Handgrip strength	32.00 (24.80–34.85)	32.00 (26.25–34.75)	2 (1.1–2.0)	0.02	−2.41
VO_2_max ^1^ mL/min/kg	16.25 (13.18–24.18)	17.55 (12.95–21.23)	0.0 (−0.5–2.5)	0.60	−0.53
VO_2_max at AT ^2^/kg	12.60 (9.65–17.15)	13.55 (9.68–17.55)	−0.25 (−3.15–1.55)	0.67	0.67
Maximum exercise capacity (Watt)	69.00 (62.00–124.00)	75.00 (55.50–143.25)	13 (6.0–27.0)	0.02	−2.37
FVC ^3^ (liter)	2.86 (2.71–4.42)	3.19 (2.66–4.07)	0.06 (−0.16–0.32)	0.48	−0.70
FEV1 ^4^ (liter)	2.32 (1.05–3.35)	2.19 (1.12–3.47)	0.03 (−0.04–0.12)	0.57	−0.56

^1^ VO_2_ Max: maximal oxygen uptake, ^2^ AT: aerobic threshold point, ^3^ FVC: forced vital capacity, ^4^ FEV1: forced expiratory volume in 1 s.

**Table 4 nutrients-13-03493-t004:** Surgical outcomes; number (percentages), median (IQR).

	Study Population (*n* = 9)
Surgery; *n* (%)	6
Left hemicolectomy	1 (16.7)
Right hemicolectomy	4 (66.7)
Sigmoidectomy	1 (16.7)
Re-admission < 30 days	0
Complication; *n* (%)	2/6 (33.3)
Microperforation	1 (16.7)
Tachycardia	1 (16.7)
Clavien Dindo, n	
II	1 (Tachycardia) (16.7)
IIIb	1 (Perforation) (16.7)
Length of hospital stay (days)	4 (3.0–8.8)
30-day mortality; n	0
1-year mortality; n	0

## Data Availability

Data are available upon request from the first author (T.T.T. Tweed).

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
