# Peer review of "Feasibility and Efficiency of the BEFORE (Better Exercise and Food, Better Recovery) Prehabilitation Program"

_nutrients, 2021, doi:10.3390/nu13103493_

Round 1
Reviewer 1 Report
Dear Authors:
The authors have carried out the study “Better Exercise and FOod, better Recovery (BEFORe-study)”. The aim of this study was to assess the feasibility of a 4-week multimodal prehabilitation program consisting of a personalized in-hospital exercise training and nutritional intervention for colorectal cancer patients aged over 64 years prior to surgery. The authors have conducted a comprehensive, rigorous, and scientifically correct study.
Some considerations need to be taken into account:
- Why did authors choose the cutoff age of over 64?
- Table 2 describes “Exercise and nutritional data for patients; n (%), median (interquartile range). and does not fully conform to what is reflected in the text (line 243). Surgical putcomes are not reflected in table 2.
- Physical outcomes are presented in table 3 (line 254)
- The description of table 3 is not reflected in the text. It would be advisable to place table 3 in the text at the end of the paragraph located between line 243-253
- The description of Figures 2-5 are not reflected in the text. It would be advisable to place these figures in the text to facilitate their reading.
- The VAS questionnaire reflected in appendix B does not present the corresponding explanatory legend (line 487)
- Table 4 is not reflected in the text. It would be advisable to place table 4 in the text at the end of the paragraph located between line 274-281
- References should be described as recommended by the style guide of the journal. Journals should be cited as Abbreviated Journal Name. Please review the reference list and follow the journal style guide (https://www.mdpi.com/journal/nutrients/instructions#preparation).
The most important limitation of the study was based on the extremely low level of study recruitment (n: 9) of which 6 patients underwent surgery. This fact not only facilitates the development of a selection bias (recruitment of patients with affinity for training) but does not allow conclusions to be drawn for clinical practice. The conclusions of the study are based in a very low recruitment rate which reduces the power of the study and does not allow the generalization of the results.
Kind regards
Reviewer 2 Report
This study is the first prospective study specifically focusing on the feasibility of a multimodal prehabilitation program including exercise training program and with dietary nutrition in patients with old-aged colorectal malignancy.
Given the current lack of evidence of preoperative preoperative rehabilitation and nutrition, this study is considered very timely.
Reviewer 3 Report
The Better Exercise and FOod, better Recovery (BEFORe-study) manuscript is very well written and the authors explain and justify in great detail most of the observations the reader might make. However, I suggest that the authors reflect and / or modify some things.
1. Title: I think the title is misleading. The authors could add a question mark at the end of the title, otherwise a more appropriate title would be "Feasibility and efficacy of a specific prehabilitation program ...".
In fact, the same authors in the discussion affirm that it is not possible to advance any hypothesis regarding the correlation between the post-operative outcomes and the applied program.
2. Considering the very small number of participants, I would recommend that you always specify that this is a pilot study.
3. line 53 - how does this study help to overcome the problem of small samples? Maybe I would avoid putting this observation at this point.
4. line 75, in the abstract it is indicated that the subjects are older than 64 years before surgery, here they are over 65. It is better to standardize the data (or better explain what is meant)
5. line 101 - I think it is more appropriate to talk about nutritional requirements than nutritional status
6. in the diagram it is indicated that the prehabilitation had been completed by all the subjects included, but in the following text it does not seem so (some did not complete the exercise program, others did not record the diets ...). I think it is necessary to review and explain.
7. line 287 - I would specify that this is a specific program (also in the course of the text); this is important when it comes to the generalizability of the results (other programs could consist in the administration of the different nutritional products, other types of exercises ...)
8. Again with reference to the generalizability of the data, the fact that the authors themselves argue that the program is too complex and that there has been little participation raises the question: was it worthwhile to carry out this study? Perhaps the authors shouls specify that the study was done to provide evidence to skeptics that this type of intervention is not that complex and that thanks to such programs important results may be reached ...
9. line 305 - the authors talk about the difficulty of traveling to the hospital and only at this point I fully understood how the study was carried out. Initially the authors were talking about in hospital programm and I believed that the patients were already hospitalized when they participated in the program (although that seemed very difficult to implement). How the program was implemented must be better explained from the begining (methodology).
10. If we talk about the results of analysis of satisfaction with the taste of the products, the products used within this specific program must be specified, otherwise that part is not relevant for readers.
11. table 4 - are we sure that 33% of complications are a low number of complications? This aspect must be treated very carefully.
Finally, if possible, I would make some hypotheses about the role of nutrition and exercise separately. What contributed the most to improving conditions, what was more feasible ....
Round 2
Reviewer 1 Report
Dear authors,
Thank you very much for reviewing, clarifying and modifying the allegations made previously. It is a very constructive work with high scientific rigor. I have communicated it to the editors of the journal.
Kind regards